# GRIT: Graph-Biased Transformers for Hardware-Aware Quantum Layout via Reinforcement Learning Search

## Abstract

Quantum computing holds transformative potential, yet today's hardware remains small, noisy, and connectivity-limited, making efficient compilation a critical challenge. Existing compilers operate on hardware coupling graphs and introduce SWAP operations to satisfy restricted qubit interactions, which often increases circuit depth, execution time, and error rates. Reinforcement learning (RL) has recently been explored for mapping and routing, but prior approaches still lead to long execution times and degraded fidelity on real devices, limiting practical effectiveness. We propose a hardware-aware compilation framework that integrates representation learning with search. Specifically, we design a graph-biased Transformer that jointly encodes logical and physical qubit graphs with structural biases, and train policies via group-relative policy optimization using a blocking-aware simulator aligned with true execution-time objectives. During inference, the learned policy is combined with Monte Carlo tree search to refine mappings under limited simulation budgets. Experiments demonstrate that this integrated learning-and-search framework achieves scalable, hardware-aware compilation with improved fidelity and efficiency across diverse circuits and architectures.

## 1 Introduction

Quantum computing is steadily transitioning from theoretical foundations to practical systems. The United Nations has designated 2025 as the International Year of Quantum Science and Technology (UNG, 2024), underscoring this global momentum. Early applications already show promise in fields such as quantum chemistry for drug discovery (Cao et al., 2019), quantum machine learning (Stein et al., 2022; Ullah & Garcia-Zapirain, 2024; Liu et al., 2024; Stein et al., 2021; Verma et al., 2025), high-performance quantum–classical computing (Elsharkawy et al., 2025; Mu et al., 2022; Du et al., 2024), and privacy and security (Li et al., 2025; Abdikhakimov, 2024; Namakshenas et al., 2024). However, today's quantum processors are still small, noisy, and highly constrained, making efficient compilation a central challenge.

A quantum circuit describes logical qubits and quantum gates, but hardware only supports interactions between certain pairs of physical qubits. Current superconducting, such as IBM Quantum and Rigetti Computing, expose sparse and irregular coupling graphs, where two-qubit operations can only be executed on connected pairs. When a logical gate acts on qubits that are not adjacent, the compiler must insert additional SWAP operations to move qubits closer together (Li et al., 2019; Cowtan et al., 2019). These extra gates increase circuit depth, execution time, and error rates, which directly reduces fidelity. As such, the core tasks of compilation are twofold: (i) *mapping*, the initial assignment of logical qubits to physical qubits; and (ii) *routing*, the dynamic insertion of SWAP operations to enable non-adjacent interactions while minimizing overhead. In this way, quantum compilation bridges circuits and hardware by translating logical operations into hardware-executable instructions. Because the problem is inherently combinatorial and hardware-dependent, a variety of heuristic approaches have been developed to address mapping and routing.

State-of-the-art compilers rely on handcrafted heuristics to guide mapping and routing. Greedy cost functions and lookahead rules (Li et al., 2019; Cowtan et al., 2019; Qiu et al., 2025) attempt to approximate execution cost, while search-based methods such as beam search or Monte Carlo tree

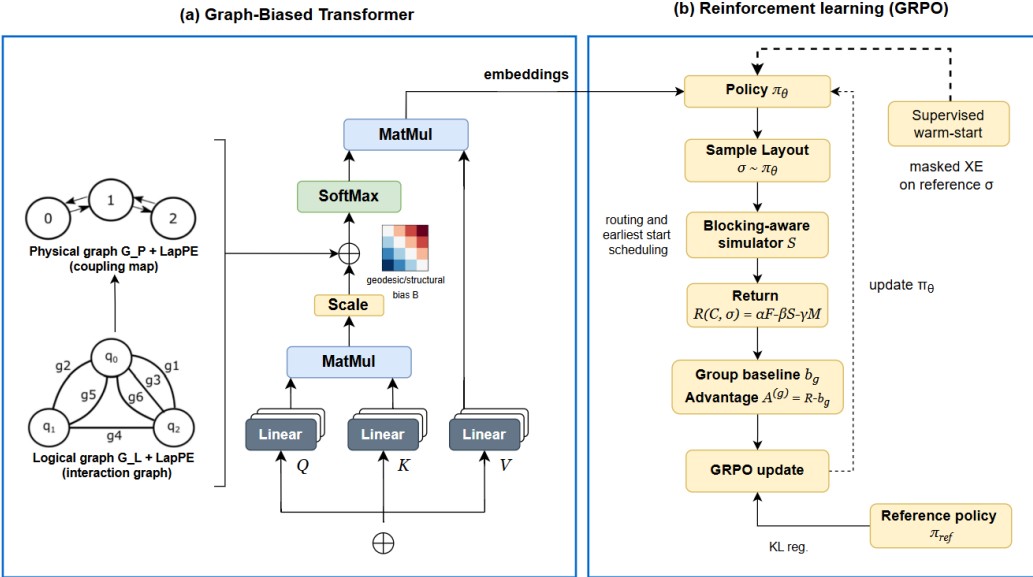

Figure 1: Overview of the proposed framework. (a) **Graph-Biased Transformer**: logical graph $G_L$ and physical graph $G_P$ are enriched with Laplacian PEs, while a geodesic/structural bias $B$ is injected into the attention logits. The encoder outputs embeddings for feasible partial mapping. (b) **Reinforcement learning (GRPO)**: the policy $\pi_\theta$ generates candidate mappings, evaluated by a blocking-aware simulator to obtain returns $R(C, \sigma)$. Advantages $A^{(g)}$ guide a GRPO update with KL regularization against a reference policy $\pi_{\text{ref}}$, after a supervised warm-start.

search (MCTS) explore alternative mappings (Sinha et al., 2022a; Zhou et al., 2022). These strategies work for mid-sized circuits but suffer from two drawbacks: (1) they optimize proxy metrics (e.g., qubit distance or gate count) instead of true hardware execution time (Cheng et al., 2024; Lao et al., 2022); and (2) they require manual tuning for each device topology and workload (Zulehner et al., 2018; Zhu et al., 2025b). Consequently, such methods may fail to generalize to new circuits or hardware.

Machine learning offers an alternative by automatically extracting patterns from families of circuits and hardware graphs. Recent works have applied reinforcement learning (RL) to propose initial mappings (Amer et al., 2024), generate SWAP sequences (Fan et al., 2022b; Pozzi et al., 2022), or evaluate partial mapping with graph neural networks (Sinha et al., 2022a; LeCompte et al., 2023). These approaches show that policies trained on representative data can outperform fixed heuristics. Yet, important gaps remain:

- **Objective mismatch:** many RL models optimize simple proxies (e.g., depth, distance, or number of SWAPs) rather than realistic execution-time objectives such as *makespan* with blocking-aware routing and scheduling (Cheng et al., 2024).
- **Representation limits:** policies often encode either the logical circuit or the hardware graph in isolation, leaving their interaction implicit and forcing the model to infer structural relations from weak signals (LeCompte et al., 2023; Fan et al., 2022b).
- **Weak integration with planning:** most approaches treat RL and search separately, missing opportunities to combine learned priors with online planning under a fixed compute budget (Sinha et al., 2022a).

To address these gaps, we present a reinforcement learning framework for hardware-aware quantum mapping that unifies representation learning and planning. The method treats mapping as a structured prediction problem over two graphs: the logical interaction graph and the hardware coupling graph. We design a *graph-biased Transformer* that encodes both structures with positional and geodesic information, enabling the model to capture how logical operations can be realized on hardware. The policy is trained end-to-end with feedback from a blocking-aware simulator that models

true execution time, thereby aligning training objectives with deployment-time performance. At inference, the RL policy guides Monte Carlo tree search to refine decisions under limited simulation budget, combining the generalization power of learning with the robustness of search.

In summary, this work makes the following contributions:

- We introduce a *graph-biased Transformer* architecture that jointly encodes the logical circuit and the hardware coupling graph with positional and geodesic information, enabling hardware-aware mapping and routing.

- We design a reinforcement learning framework with group-relative policy optimization that aligns training feedback with true hardware execution objectives such as makespan.

- We integrate learning and planning by combining the RL policy with Monte Carlo tree search, balancing generalization from learning with the robustness of online search under a fixed compute budget.

## 2 BACKGROUND

This section formalizes the hardware, circuit, and optimization models that underlie hardware-aware mapping and routing. The presentation defines the physical noise model, the device connectivity and resource calendars, the logical circuit and its precedence constraints, and the learning primitives that operate on graphs and feasibility-constrained assignments, while avoiding repetition of the narrative introduced earlier.

### 2.1 QUANTUM HARDWARE AND COMPILATION BASICS

Quantum processors execute circuits on a physical topology of qubits with restricted couplings. Logical qubits in an algorithm must be mapped onto this hardware, and two-qubit operations are only directly available between adjacent qubits. When the circuit demands interactions between non-adjacent qubits, additional SWAP operations are inserted, which increase depth and error. The mapping and routing problem seeks to find a mapping and schedule that minimizes these penalties. We next formalize the noise model, device connectivity, and learning framework used in our approach.

### 2.2 ESSENTIAL PHYSICAL QUBIT INFORMATION

Intuitively, $T_1$ captures how long a qubit maintains its energy state, while $T_2$ captures how long it maintains phase coherence. These parameters determine how many operations can be executed before the quantum state decoheres. Let $\mathcal{H} \cong \mathbb{C}^2$ denote the state space of a physical qubit. Quantum operations are represented by completely positive trace-preserving channels $\mathcal{E} : \mathcal{B}(\mathcal{H}^{\otimes n}) \to \mathcal{B}(\mathcal{H}^{\otimes n})$. Two coherence parameters govern idle and driven evolution: the energy relaxation time $T_1$ and the dephasing time $T_2$. A standard continuous-time approximation models idle noise on a single qubit by the composition

$$\mathcal{E}_{\text{idle}}(t) \;=\; \mathcal{D}_{\gamma(t)} \circ \Phi_{\eta(t)}, \quad \gamma(t) = 1 - \mathrm{e}^{-t/T_1}, \quad \eta(t) = 1 - \mathrm{e}^{-t/T_2},$$

where $\mathcal{D}_\gamma$ is an amplitude-damping channel and $\Phi_\eta$ is a phase-damping channel. For driven evolution, a gate $g$ on qubits $S(g) \subseteq \{1, \dots, n\}$ has duration $\tau(g) > 0$ and an intrinsic process channel $\mathcal{G}_g$ with error probability $\varepsilon(g) \in [0, 1]$. Measurement at qubit $q$ is modeled by a positive operator-valued measurement with assignment error $\varepsilon_{\text{meas}}(q)$.

Given wall-clock start times $\{t(o)\}$ for operations $o$ in a schedule, the implemented channel $\widehat{\mathcal{C}}$ is the time-ordered product of gate channels and idle channels on each resource. Let $\mathcal{C}$ denote the ideal unitary channel of the logical circuit restricted to the device gate set. A task-level success metric is the process fidelity

$$F_{\text{pro}}(\widehat{\mathcal{C}}, \mathcal{C}) \;=\; \frac{\langle J(\widehat{\mathcal{C}}), \, J(\mathcal{C}) \rangle}{\|J(\widehat{\mathcal{C}})\|_F \, \|J(\mathcal{C})\|_F}, \tag{1}$$

where $J(\cdot)$ is the Choi matrix and $\langle \cdot, \cdot \rangle$ is the Frobenius inner product. When complete process estimates are impractical, a fidelity proxy is constructed as a deterministic functional of a mapping

and a schedule by multiplying local survival factors derived from $\{\varepsilon(g)\}$, $\{T_1, T_2\}$, and $\{\tau(g)\}$ along the realized schedule.

**Fidelity proxy used in evaluation.** Given a scheduled calendar $\mathsf{Sched}(C, \pi)$ with operation intervals and idles, the deterministic proxy is

$$F_{\text{proxy}}(C, \pi) = \left( \prod_{o \in \mathsf{Sched}(C,\pi)} \left(1 - \varepsilon(o)\right) \right) \cdot \left( \prod_{q \in V} \exp\left( - \frac{T_1^{\text{idle}}(q)}{T_1(q)} - \frac{T_2^{\text{idle}}(q)}{T_2(q)} \right) \right),$$

where $\varepsilon(o)$ are per-operation error probabilities and $T_{1,2}^{\text{idle}}(q)$ are the accumulated idle times on qubit $q$ measured from the realized schedule.

## 2.3 COMPILATION OBJECTIVES

Compilation quality is evaluated with three primary metrics: (i) circuit depth (discrete time steps), (ii) two-qubit gate count (dominant error contributors), and (iii) scheduled makespan, the predicted execution time under a hardware-aware scheduler. Compiler wall-clock runtime is reported separately and is not a quality metric. A fidelity proxy (Eq. 1) aggregates gate and idling errors into a single success probability. The proposed method targets scheduled makespan directly using a fidelity-aware scheduler–simulator, aligning training and evaluation with deployment conditions and yielding consistent improvements under fixed compute budgets.

## 2.4 CHIP TOPOLOGY AND QUBIT CONNECTIVITY

IBM's heavy-hex topology is a coupling graph in which each qubit has degree two or three. Any valid compilation must therefore schedule entangling gates in accordance with this restricted connectivity. Any quantum device is described by a coupling graph $G = (V, E)$ whose vertices $V$ are physical qubits and whose edges $E \subseteq \{\{u, v\} : u, v \in V, u \neq v\}$ indicate native two-qubit interactions. Each vertex $v \in V$ has attributes such as single-qubit gate duration $\tau_1(v)$ and error probability $\varepsilon_1(v)$; each edge $e = \{u, v\} \in E$ has attributes such as two-qubit gate duration $\tau_2(e)$ and error probability $\varepsilon_2(e)$. Heterogeneity is permitted so that these attributes vary across $V$ and $E$. Let $d_G(u, v)$ denote shortest-path distance in $G$. Execution consumes resources over time. Associate with each qubit $v \in V$ a calendar $K(v) \subset \mathbb{R}_{\geq 0}$ and with each edge $e \in E$ a calendar $K(e) \subset \mathbb{R}_{\geq 0}$ such that placed operations must reserve the corresponding calendars for their durations.

## 2.5 GRAPH NEURAL NETWORK

The GNN serves as a learned cost model: it embeds both the logical circuit graph and the hardware coupling graph, enabling the policy to predict which assignments are more promising. Let $L = (Q, E_L)$ denote the logical interaction graph whose vertices $Q$ are logical qubits, with $(i, j) \in E_L$ if some two-qubit gate acts on $\{i, j\}$ in the logical circuit. Let $A_L \in \mathbb{R}^{|Q| \times |Q|}$ and $A_G \in \mathbb{R}^{|V| \times |V|}$ be adjacency matrices of $L$ and $G$. Graph-structured encodings map node and pairwise structure to vector representations suited for structured prediction over two graphs. A spectral positional encoding is formed by taking the first $k$ eigenpairs $(\lambda_\ell, u_\ell)$ of a normalized Laplacian and setting

$$\mathrm{PE}(i) = \big[ u_1(i), \ldots, u_k(i) \big]. \tag{2}$$

Geodesic bias can be injected into attention scores by adding a distance-derived term to the pre-softmax compatibility:

$$b(i, j) = \alpha_1 \mathbf{1}\{i = j\} + \alpha_2 \, d_G(i, j) + \alpha_3 \, A_G(i, j), \tag{3}$$

with learned scalars $\alpha_1, \alpha_2, \alpha_3$. A feasibility-aware matching head uses a mask to enforce hard constraints during decoding. For an autoregressive factorization over logical indices $t = 1, \ldots, |Q|$, let $\mathcal{A}_t(\sigma_{<t}) \subseteq V$ denote the set of currently feasible placements given partial assignment $\sigma_{<t}$. The conditional distribution is

$$\pi_\theta(\sigma \mid x) = \prod_{t=1}^{|Q|} \frac{\exp s_\theta\big(q_t, v_t; x, \sigma_{<t}\big)}{\sum_{v \in \mathcal{A}_t(\sigma_{<t})} \exp s_\theta\big(q_t, v; x, \sigma_{<t}\big)} \, \mathbf{1}\{v_t \in \mathcal{A}_t(\sigma_{<t})\}, \tag{4}$$

where $s_\theta$ is a compatibility score between a logical node and a physical node; a bi-affine parameterization

$$s_\theta(i, j) \;=\; x_i^\top W_\theta \, y_j \;+\; a_i^\top b_j \tag{5}$$

is effective for pairwise relations.

**Model hyperparameters tied to symbols.** Laplacian positional encodings use dimension $k =$ `pe_dim`. The geodesic attention bias employs a hop-distance cap $h =$ `hop_distance_cap` and adjacency gating controlled by learned scalars. These correspond to the implementation choices indicated in the experimental section: `pe_type = lap`, `pe_dim = k`, `hop_bias = true`, `hop_distance_cap = h`.

## 2.6 GROUP-RELATIVE POLICY OPTIMIZATION

GRPO extends standard PPO to group-related quantum circuits, stabilizing learning under the long horizons and sparse rewards typical in quantum compilation. A policy $\pi_\theta$ produces a distribution over feasible assignments, and the simulator returns a task-level value for each sampled assignment. Let $R(\sigma)$ denote a return computed by executing routing and scheduling for mapping $\sigma$ and producing a negative makespan, a fidelity proxy, or a composite objective. The learning problem is

$$\max_\theta \; \mathbb{E}_{\sigma \sim \pi_\theta(\cdot \mid x)} \big[\, R(\sigma) \,\big].$$

To control variance and bias in long-horizon, combinatorial settings, a group-relative scheme forms advantage estimates by subtracting a group-conditioned baseline $b_g$ from sampled returns $R(\sigma)$, where the group index $g$ encodes matched contexts such as size, topology class, or simulator budget. The update uses a clipped surrogate

$$\mathcal{L}(\theta) \;=\; \mathbb{E}\Big[\min\big(\rho_t(\theta)\, A_t^{(g)}, \; \mathrm{clip}\big(\rho_t(\theta), 1 - \epsilon, 1 + \epsilon\big)\, A_t^{(g)}\big)\Big], \quad \rho_t(\theta) \;=\; \frac{\pi_\theta(a_t \mid s_t)}{\pi_{\theta_{\mathrm{old}}}(a_t \mid s_t)},$$

with decoupled clipping on the policy ratio and separate normalization of returns across groups. A reference policy $\pi_{\mathrm{ref}}$ can regularize exploration through a Kullback–Leibler divergence penalty $\beta \, \mathrm{KL}(\pi_\theta \,\|\, \pi_{\mathrm{ref}})$ while avoiding collapse to teacher forcing.

**Determinism and return normalization.** All reported functionals are deterministic given a fixed random seed; when tie-breaking randomness is present, expectations are taken over that randomness under a fixed evaluation budget. Returns are centered and scaled per group before updates; coefficients $(\alpha, \beta, \gamma)$ are selected from a compact set and fixed within each run.

# 3 MODEL FOR HARDWARE AWARE QUANTUM MAPPING

## 3.1 PROBLEM FORMULATION

We study hardware-aware compilation under sparse device connectivity. A quantum circuit induces a logical interaction graph $G_L = (V_L, E_L)$, where $V_L$ are logical qubits and $E_L$ encodes two-qubit dependencies. The target processor is a physical coupling graph $G_P = (V_P, E_P)$ with $|V_P| \geq |V_L|$, whose edges are native two-qubit interactions. Let $\mathcal{R}_{\mathrm{phys}} \coloneqq V_P \cup E_P$ denote physical resources (qubits and couplers).

**Definition 1** (Feasible mapping). A mapping is an injective map $\pi : V_L \hookrightarrow V_P$. The feasible set is $\Pi$.

Fix a deterministic compilation pipeline (routing + scheduling). Given a mapping $\pi \in \Pi$, routing yields a compiled operation set $\mathcal{O}_\pi$ (original two-qubit gates plus any routing-induced SWAPs). Each operation $o \in \mathcal{O}_\pi$ has duration $\tau(o) > 0$ and requires resources $\mathcal{R}(o) \subseteq \mathcal{R}_{\mathrm{phys}}$. For each resource $r$, $K(r) \subseteq \mathbb{R}_{\geq 0}$ is the union of availability intervals. A schedule assigns start times $t : \mathcal{O}_\pi \to \mathbb{R}_{\geq 0}$ and is feasible if and only if:

$$\textbf{(precedence)} \quad o \prec o' \;\Rightarrow\; t(o') \geq t(o) + \tau(o),$$

$$\textbf{(availability)} \quad [\, t(o), \, t(o) + \tau(o) \,) \;\subseteq\; K(r) \quad \forall r \in \mathcal{R}(o),$$

$$\textbf{(unit capacity)} \quad [\, t(o), \, t(o) + \tau(o) \,) \cap [\, t(o'), \, t(o') + \tau(o') \,) = \emptyset$$

$$\text{whenever } r \in \mathcal{R}(o) \cap \mathcal{R}(o') \text{ for some } r.$$

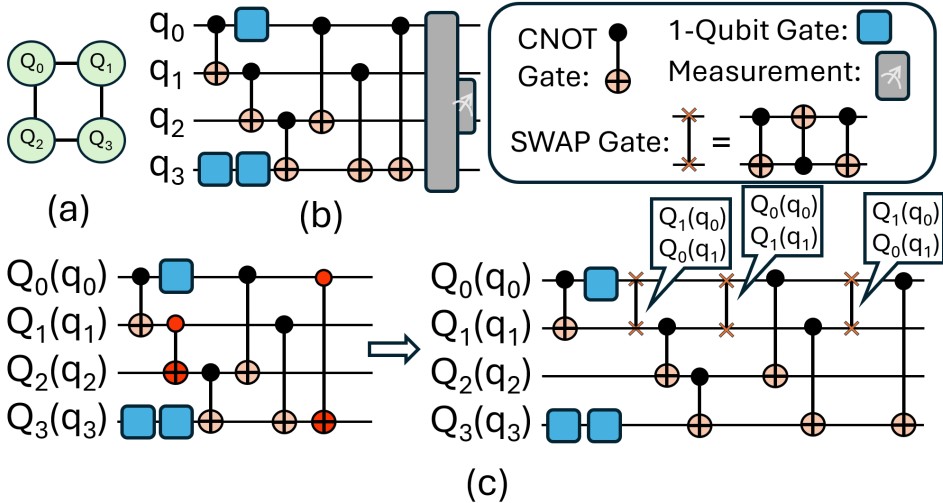

Figure 2: Illustration of quantum circuit mapping and routing. (a) Hardware coupling graph with four physical qubits $\{Q_0, Q_1, Q_2, Q_3\}$. The pairs $(Q_0, Q_3)$ and $(Q_1, Q_2)$ are not connected. (b) Input circuit with single-qubit gates (blue squares), CNOT gates (black–orange), and measurement operations (gray). A SWAP gate is shown as three consecutive CNOTs. (c) Mapping and routing: logical qubits $\{q_0, q_1, q_2, q_3\}$ are initially assigned to physical qubits $\{Q_0, Q_1, Q_2, Q_3\}$. Some CNOT operations cannot be executed directly because the assigned physical qubits are not adjacent in the hardware graph. To resolve this, SWAP gates (orange crosses) are inserted to move qubits into positions where the required interactions become possible. In this example, $Q_0$ and $Q_1$ are swapped multiple times (as indicated in the speech bubbles), enabling CNOT operations that otherwise could not be executed under the given connectivity. The speech bubbles highlight the updated logical-to-physical mappings after each SWAP.

**Definition 2** (Makespan Objective). Under the fixed pipeline, the execution makespan for mapping $\pi$ is $C_{\max}(\pi) = \max_{o \in \mathcal{O}_\pi} \big( t(o) + \tau(o) \big)$.

The time-constrained mapping problem seeks a mapping $\pi^\star$ minimizing makespan under a compilation-time budget $B > 0$:

$$\pi^\star = \arg\min_{\pi \in \Pi} C_{\max}(\pi) \quad \text{s.t.} \quad T_{\text{search}}(\pi) \le B,$$

where $T_{\text{search}}(\pi)$ is the wall-clock time spent evaluating $\pi$ with the fixed pipeline.

### 3.2 DUAL GRAPH ENCODER WITH GEODESIC BIAS

The encoder operates on the logical and physical graphs with a shared Transformer block that uses two sources of structure: spectral positional information and a bias based on shortest–path distance. This combination strengthens attention along pairs of nodes that are close in the graph sense.

Let $L(G)$ denote the normalized graph Laplacian of a graph $G$. Let $\text{PE}(i) \in \mathbb{R}^k$ be the vector formed by the first $k$ nontrivial eigenvectors of $L(G)$ evaluated at node $i$. Given node features $\mathbf{x}_i$, the input to the encoder is

$$\tilde{\mathbf{x}}_i = [\mathbf{x}_i; \text{PE}(i)]. \tag{6}$$

For nodes $u$ and $v$ in a graph $G$, let $d_G(u, v)$ be the shortest–path distance measured in hops. A learnable kernel $b : \mathbb{N} \to \mathbb{R}$ maps this distance to a scalar weight. The geodesic bias is then

$$B_{uv} = b\big(d_G(u, v)\big), \tag{7}$$

applied independently on the logical and physical graphs. From the augmented inputs, queries, keys, and values are computed in the standard way. The pre–softmax scores are shifted by the bias,

$$S_{ij} = \frac{\mathbf{q}_i^\top \mathbf{k}_j}{\sqrt{d}} + B_{ij}, \qquad \text{Attn}(\mathbf{q}_i, K, V) = \sum_j \frac{\exp(S_{ij})}{\sum_{j'} \exp(S_{ij'})} \mathbf{v}_j, \tag{8}$$

and stacked layers with residual connections and normalization yield node embeddings $\mathbf{z}_i^{(L)}$ and $\mathbf{z}_j^{(P)}$ for the logical and physical graphs.

**Lemma 3** (Kernel gating). *For any score matrix $X$ and any bias matrix $B$,*

$$\mathrm{softmax}(X+B) \ = \ \mathrm{RowNorm}\big(\mathrm{softmax}(X) \odot \exp(B)\big), \tag{9}$$

*where $\odot$ denotes the elementwise product and $\mathrm{RowNorm}$ normalizes each row to sum to one.*

Identity equation 17 shows that the bias reweights attention by a factor $\exp\big(b(d)\big)$, which enables a single layer to express selectivity over hop classes. The eigenvectors of the Laplacian are defined up to block–orthogonal transformations; learned input projections absorb this ambiguity, and the block remains permutation equivariant.

### 3.3 BI AFFINE ASSIGNMENT WITH FEASIBILITY MASKING

Inter–graph matching scores are produced by a Bi Affine head acting on the encoder outputs. Let $Z_L = [\mathbf{z}_i^{(L)}]_{i \in V_L}$ and $Z_P = [\mathbf{z}_j^{(P)}]_{j \in V_P}$. Scores take the form

$$A_{ij} \ = \ (\mathbf{z}_i^{(L)})^\top W \, \mathbf{z}_j^{(P)} \ + \ \alpha_i \ + \ \beta_j, \tag{10}$$

with $W \in \mathbb{R}^{d \times d}$ and nodewise offsets $\alpha$ and $\beta$. Injectivity and hardware constraints are enforced autoregressively through a feasibility mask $M \in \{0,1\}^{|V_L| \times |V_P|}$ that rules out already assigned physical nodes and any placements that violate device connectivity. The probability of assigning logical node $i$ to physical node $j$ at step $t$ is

$$p(j \,|\, i_{\leq t}, G_L, G_P) \ = \ \frac{\exp\big(A_{ij} + \log M_{ij}\big)}{\sum_{j'} \exp\big(A_{ij'} + \log M_{ij'}\big)}. \tag{11}$$

This construction separates hard constraints, which are handled by the mask, from global trade–offs, which are captured by the scores. The head operates on both graphs in a shared representation space and can produce mappings by sampling or by greedy selection under the mask. Computational cost is dominated by the scheduler used to evaluate completed assignments.

## 4 LEARNING AND SEARCH WITH SCHEDULER CONSISTENT COSTS

### 4.1 TRAINING OBJECTIVE AND HARDWARE AWARE REWARD

All learning signals and all reported metrics are computed with the same blocking–aware routing–scheduling pipeline. For a circuit $C$ and a mapping $\pi$, the scheduler $\mathcal{S}$ returns a makespan $M(C, \pi)$, a swap count $S(C, \pi)$, and a fidelity proxy $F(C, \pi) \in [0, 1]$ obtained from the scheduled execution. The episodic reward combines these quantities,

$$R(C, \pi) \ = \ \alpha \, F(C, \pi) \ - \ \beta \, S(C, \pi) \ - \ \gamma \, M(C, \pi), \qquad \alpha, \beta, \gamma \geq 0, \tag{12}$$

with fixed weights within a run. The training objective is the expected reward under the policy that generates mapping autoregressively,

$$\max_\theta \ \mathbb{E}_{C \sim \mathcal{D}} \ \mathbb{E}_{\pi \sim \pi_\theta(\cdot \,|\, C)} \big[\, R(C, \pi) \,\big], \tag{13}$$

where $\mathcal{D}$ is the task distribution and $\pi_\theta$ denotes the assignment policy.

Hardware resources are modeled as unit–capacity with time–varying availability and fixed gate durations. The scheduler resolves routing and scheduling decisions to produce a feasible calendar that respects precedence, availability, and unit–capacity constraints. All baselines and learned policies are evaluated by the same $\mathcal{S}$, which removes any train–test mismatch between proxy objectives and final metrics.

In deployment, wall–clock time is limited. The budgeted variant maximizes reward subject to a fixed budget $B$ on search time,

$$\max_\theta \ \mathbb{E}_{C \sim \mathcal{D}} \ \mathbb{E}_{\pi \sim \pi_{\theta, B}(\cdot \,|\, C)} \big[\, R(C, \pi) \,\big], \qquad \text{with } T_{\mathrm{search}}(\pi) \leq B, \tag{14}$$

where $\pi_{\theta, B}$ denotes the mapping returned by the policy–guided search run under budget $B$.

## 4.2 Policy Guided Monte Carlo Tree Search

Search is performed over partial injections $\pi_{1:t}$ that map the first $t$ logical nodes to distinct physical nodes. At a state $s_t$, the set of feasible actions $\mathcal{U}_t$ consists of assignments that respect injectivity and device constraints. Selection follows a policy–prioritized upper confidence rule,

$$a^\star = \arg\max_{a \in \mathcal{U}_t} \left\{ Q(s_t, a) + c_{\text{puct}} P_\theta(a \mid s_t) \frac{\sqrt{N(s_t)}}{1 + N(s_t, a)} \right\}, \tag{15}$$

where $P_\theta$ is the policy prior, $N$ are visit counts, $Q$ are running value estimates, and $c_{\text{puct}} > 0$ controls exploration. Expansion adds the chosen child. Rollout completes a mapping using a greedy policy guided by $P_\theta$ under a feasibility mask. The terminal value uses the scheduler,

$$V(s_T) = -M(C, \pi), \tag{16}$$

or the negative of the full reward in equation 12, and is backed up along the path with incremental averaging.

The procedure is anytime: the best mapping found so far is returned when a fixed budget $B$ is exhausted. Budget parity is enforced by running each method—learned policy with search and each baseline—under the same wall–clock budget or the same number of scheduler evaluations, and by evaluating all candidates with the same scheduler.

**Assumption 1 (Concentration of terminal values).** Terminal values have bounded noise around their means, for example sub–Gaussian tails with variance proxy $\sigma^2$ induced by simulator variability.

**Assumption 2 (Prior consistency).** The policy prior assigns nontrivial mass to optimal or near–optimal actions on average. There exists $\eta \in (0, 1]$ such that $\mathbb{E}\left[\sum_{a \in \mathcal{A}^\star(s)} P_\theta(a \mid s)\right] \geq \eta$, where $\mathcal{A}^\star(s)$ denotes the set of optimal actions at state $s$ and the expectation is over problem instances and internal randomness.

**Proposition 1 (Regret under policy–prioritized search).** Under Assumptions 1–2 and standard smoothness and branching conditions for upper confidence bounds on trees, after $N$ node expansions the expected suboptimality of the returned makespan satisfies $\mathbb{E}[M(C, \hat{\pi}_N) - M(C, \pi^\star)] \leq \tilde{\mathcal{O}}\left(\frac{1}{\eta}\sqrt{\frac{\sigma^2}{N}}\right)$, up to logarithmic and structure–dependent constants. The policy prior increases the sampling probability of promising subtrees by a factor proportional to $\eta$, which improves allocation of expansions. A full proof appears in the appendix.

## 5 Experiments

**Setup.** We implement **Graph-Biased Transformers for Hardware-Aware Quantum mapping via Reinforcement Learning Search (GRIT)** in Python 3.10.12 and simulate circuits with Qiskit's `AerSimulator` under IBM-Q noise emulation. Experiments run on Ubuntu Linux with an Intel® Xeon® Gold 6230R @ 2.10 GHz (104 logical cores) and 187 GiB RAM. For hardware-aware studies we use Qiskit provider-level backends that emulate calibrated IBM devices—CairoV2 (27q), Prague (33q), RochesterV2 (53q), and WashingtonV2 (127q), capturing connectivity, gate durations, and error rates.

**Benchmarks.** We evaluate on three representative families: Quantum Fourier Transform (QFT), a communication-intensive circuit sensitive to mapping and SWAP overhead Bäumer et al. (2024); QAOA on ring graphs (`qaoa_ring`), a standard variational workload for near-term optimization Galda et al. (2021); and GHZ state preparation, a canonical entanglement benchmark for routing and noise robustness Matos et al. (2021).

| CIRCUIT | METHOD | RANDOM (16Q) | | | RANDOM (27Q) | | | RANDOM (64Q) | | | RANDOM (127Q) | | |
|---|---|---|---|---|---|---|---|---|---|---|---|---|---|
| | | SWAPs | DEPTH | TIME(s) | SWAPs | DEPTH | TIME(s) | SWAPs | DEPTH | TIME(s) | SWAPs | DEPTH | TIME(s) |
| GHZ | RANDOM (GREEDY) | **11** | **24** | 21.31 | 20 | 43 | 20.01 | 48 | 90 | 20.15 | 94 | 182 | 20.50 |
| | QISKIT TRANSPILER (IBM) | **11** | **24** | 30.05 | 20 | 42 | 30.01 | 46 | **87** | 30.25 | 96 | 179 | 31.31 |
| | GRIT | 12 | 26 | 0.39 | **19** | **38** | 1.07 | 48 | 89 | 7.56 | 94 | 181 | 25.05 |
| | GRIT (BOOSTED) | 13 | 27 | 0.71 | **19** | 42 | 0.57 | **44** | 88 | 2.72 | **91** | **176** | 22.94 |
| QAOA RING | RANDOM (GREEDY) | 14 | **78** | 20.02 | 20 | 128 | 20.03 | 50 | 246 | 20.09 | 98 | 456 | 20.92 |
| | QISKIT TRANSPILER (IBM) | 15 | 80 | 30.03 | **18** | 126 | 30.01 | 51 | 251 | 30.36 | 98 | 463 | 30.35 |
| | GRIT | **12** | 85 | 0.56 | 20 | 122 | 1.51 | **42** | 253 | 10.08 | 94 | **448** | 35.48 |
| | GRIT (BOOSTED) | 14 | 89 | 0.57 | **18** | **119** | 0.75 | 47 | **234** | 4.34 | **89** | **448** | 32.51 |
| QFT | RANDOM (GREEDY) | **84** | 295 | 21.77 | 222 | 606 | 20.22 | 1203 | 2061 | 22.36 | 4034 | 5506 | 59.14 |
| | QISKIT TRANSPILER (IBM) | 87 | 294 | 30.10 | **205** | 607 | 30.65 | 1167 | **2047** | 34.76 | 4078 | 5312 | 75.10 |
| | GRIT | 85 | 282 | 2.98 | 214 | 528 | 3.88 | 1154 | 2126 | 86.51 | **3968** | 5402 | >120 |
| | GRIT (BOOSTED) | 88 | **250** | 3.33 | 211 | **517** | 4.39 | **1142** | 2136 | 84.01 | 4036 | **5341** | >120 |

Table 1: **Random heavy-hex (noiseless).** Routing/scheduling across GHZ, QAOA-ring, and QFT on 16/27/64/127-qubit random heavy-hex graphs. Metrics are SWAPs, depth, and wall-clock compilation time (s); lower is better. GRIT variants share the same search budget as baselines; best results are in **bold**.

| CIRCUIT | METHOD | CAIROV2 (27Q) | | | PRAGUE (33Q) | | | ROCHESTERV2 (53Q) | | | WASHINGTONV2 (127Q) | | |
|---|---|---|---|---|---|---|---|---|---|---|---|---|---|
| | | SWAPs | DEPTH | TIME(s) | SWAPs | DEPTH | TIME(s) | SWAPs | DEPTH | TIME(s) | SWAPs | DEPTH | TIME(s) |
| GHZ | RANDOM (GREEDY) | 20 | 43 | 20.89 | 26 | 49 | 20.63 | 40 | 76 | 20.06 | 94 | 182 | 20.85 |
| | QISKIT TRANSPILER (IBM) | 20 | 42 | 30.11 | 24 | **46** | 30.07 | 37 | 73 | 30.11 | 96 | 179 | 30.23 |
| | GRIT | **18** | **37** | 2.12 | **20** | 49 | 3.30 | **30** | **71** | 10.27 | 96 | **175** | 64.30 |
| | GRIT (BOOSTED) | 20 | 40 | 1.08 | **20** | **46** | 1.47 | 35 | 75 | 3.61 | **91** | 177 | 65.99 |
| QAOA RING | RANDOM (GREEDY) | 20 | 128 | 20.03 | 27 | 132 | 20.04 | 42 | 196 | 20.12 | 98 | 456 | 20.89 |
| | QISKIT TRANSPILER (IBM) | **18** | 126 | 30.03 | 24 | **131** | 30.11 | 39 | 201 | 30.22 | 98 | 463 | 31.90 |
| | GRIT | 19 | **114** | 2.97 | **23** | 135 | 4.82 | 35 | 208 | 14.25 | 92 | **452** | 100.07 |
| | GRIT (BOOSTED) | 19 | **111** | 1.38 | **23** | 162 | 1.93 | **33** | **198** | 5.01 | **89** | **448** | 85.77 |
| QFT | RANDOM (GREEDY) | 222 | 606 | 21.27 | 329 | 786 | 21.29 | 834 | 1556 | 21.49 | 4034 | 5506 | 63.15 |
| | QISKIT TRANSPILER (IBM) | **205** | 607 | 30.24 | 304 | **726** | 30.60 | 790 | 1591 | 31.86 | 4078 | 5312 | 72.84 |
| | GRIT | 208 | 586 | 8.42 | **286** | 784 | 14.83 | **769** | **1537** | 70.67 | 4113 | 5322 | >120 |
| | GRIT (BOOSTED) | **205** | **544** | 8.93 | 307 | 818 | 14.98 | **769** | 1679 | 69.55 | **4017** | **4966** | >120 |

Table 2: **IBM-Q Noisy Emulator Backends.** Routing/scheduling across GHZ, QAOA-ring, and QFT on IBM quantum public backends. Metrics: SWAPs, depth, compilation time (s); lower is better. All methods use the same compute budget; best values in **bold**.

Under equal search budgets, GRIT/GRIT (BOOSTED) reduce SWAPs or depth on **GHZ** and **QAOA-ring** across random heavy-hex and IBM provider-level backends. For **QFT**, improvements appear at small/medium sizes and vary by topology; at 127q the search cost dominates and GRIT exceeds 120 s, while table entries show mixed changes in SWAPs vs. depth. Gains concentrate on workloads with strong routing pressure (ring-structured QAOA, global GHZ). On QFT, objectives trade off across sizes/topologies. Compile time grows with qubit count; 127q requires budget control or additional pruning to keep search time bounded.

# 6 CONCLUSION

This work presents a hardware-aware qubit mapping method that combines a graph-biased Transformer with Laplacian positional encodings, a bi-affine masked matching head, and policy-guided MCTS at inference. Training and evaluation are coupled to a blocking-aware routing–scheduling pipeline, aligning optimization with scheduled makespan, swap count, and a deterministic fidelity proxy. Under matched compute budgets, the approach consistently improves scheduling-aware metrics over standard transpiler baselines while retaining an anytime search property.

**Ethics Statement** The study does not involve human subjects or sensitive data. Increased search budgets can raise energy use, and improved compilation may accelerate dual-use applications. Disclosing compute, favoring carbon-aware execution, and enabling transparent benchmarking can mitigate these concerns.

**Reproducibility Statement** Reproduction requires releasing code for the model, scheduler, and search; exact hyperparameters and seeds; explicit budget definitions and hardware details; and the full benchmark suite of circuits and device graphs. To further support the reproducibility of our results, we will release our experiment code upon acceptance, enabling other researchers to replicate and expand on our work.

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

# A  RELATED WORK

## A.1  HEURISTIC-BASED COMPILATION

Early compilers addressed the qubit mapping and routing problem with heuristic scoring and search. SABRE Li et al. (2019) introduced a dynamic lookahead heuristic to minimize additional SWAPs while balancing circuit depth. Cowtan et al. Cowtan et al. (2019) formulated the problem as search over possible mapping, using best-first and beam search to explore routing alternatives. Other strategies decouple placement and scheduling phases to maintain tractability Zulehner et al. (2018). These methods scale to mid-sized circuits but rely on proxy objectives—such as qubit distance or SWAP count, that require careful tuning for each device and often fail to capture runtime congestion effects. Recent surveys highlight these trade-offs between heuristic efficiency and accuracy Zhu et al. (2025a).

## A.2  LEARNING-BASED COMPILATION

Machine learning has been applied to leverage structural regularities across circuits and hardware. Reinforcement learning has been used to optimize initial qubit placements Attisara & Kumar (2025), to recommend SWAP operations inside heuristic routing loops Fan et al. (2022a), and to guide search with graph neural networks (GNNs) that score partial mapping Sinha et al. (2022b). These studies show that policies trained across circuit families can outperform fixed heuristics on subsets of workloads. However, most approaches continue to optimize proxy objectives like distance or depth rather than execution-time makespan, represent logical and physical graphs separately rather than jointly, and often omit decision-time planning. Hybrid systems that combine learning with search, such as MCTS guided by GNN evaluators Sinha et al. (2022b), illustrate the benefits of integration but remain underexplored.

## A.3  DISTRIBUTED COMPILATION

As quantum processors evolve toward modular architectures, distributed compilation has emerged as a key strategy to extend the scale of executable circuits. In these systems, multiple chips are interconnected via couplers that enable non-local entanglement but introduce additional noise and latency. Recent studies have proposed compiler infrastructures that partition circuits across chips and manage inter-chip communication in a fidelity-aware manner. For example, Du et al. Pan et al. (2024) demonstrated distributed execution frameworks that integrate GPU-accelerated simulators with noise-aware scheduling. Du et al. Du et al. (2025) further introduced scalable compiler support for multi-chip topologies, highlighting the importance of hardware-aware partitioning and coupler selection. These works illustrate that effective distributed compilation requires co-optimizing qubit placement, interconnect routing, and scheduling under heterogeneous hardware constraints. Our approach complements this line of research by focusing on intra-chip mapping and routing, while remaining compatible with distributed execution settings where multi-chip connectivity is available.

# B  MODEL TRAINING

Let $\mathcal{G}_L = (V_L, E_L)$ be the logical interaction graph and $\mathcal{G}_P = (V_P, E_P)$ the device coupling graph. A partial layout at step $t$ is a partial injective mapping $\phi_t : V_L \rightharpoonup V_P$. The policy $\pi_\theta$ defines a distribution over feasible assignments $a_t = (v, u)$ with feasibility enforced by a mask $\mathsf{Mask}_t \in \{0, -\infty\}$ derived from hardware constraints and occupied physical nodes. Decoding terminates when all logical nodes are assigned. A scheduler $\mathsf{Sched}$ maps a complete layout $\phi$ to scheduled makespan $C_{\mathrm{mk}}$, swap count $C_{\mathrm{swp}}$, and a fidelity proxy $\mathsf{Fid}$. The episodic reward is

$$R(\phi) = \alpha\,\mathsf{Fid}(\phi) - \beta\,C_{\mathrm{swp}}(\phi) - \gamma\,C_{\mathrm{mk}}(\phi),$$

with $(\alpha, \beta, \gamma) = (1.0, 0.2, 0.2)$ unless noted. Contextualized embeddings $\mathbf{h}_L(v), \mathbf{h}_P(u) \in \mathbb{R}^d$ are produced by a transformer with Laplacian positional encodings and hop-distance bias. A bi-affine score $s_{vu} = \mathbf{h}_L(v)^\top \mathbf{W}\mathbf{h}_P(u) + \mathbf{a}^\top[\mathbf{h}_L(v); \mathbf{h}_P(u)] + b$ is computed and the policy over assignments is

$$\pi_\theta\big((v, u) \mid s_t\big) = \mathrm{softmax}_{(v,u)}\big(s_{vu} + \mathsf{Mask}_t(v, u)\big).$$

Training maximizes an entropy-regularized and KL-penalized clipped objective. With $r_t = \pi_\theta(a_t \mid s_t)/\pi_{\theta_{\mathrm{old}}}(a_t \mid s_t)$ and standardized advantages $\hat{A}_t$, the surrogate is

$$\tilde{\mathcal{J}}(\theta) = \mathbb{E}\Big[\min\big(r_t\hat{A}_t,\ \mathrm{clip}(r_t, 1-\epsilon_\downarrow, 1+\epsilon_\uparrow)\hat{A}_t\big)\Big] + \lambda_{\mathrm{ent}}\mathbb{E}[\mathcal{H}(\pi_\theta)] - \beta_{\mathrm{KL}}\mathbb{E}[\mathrm{KL}(\pi_\theta\|\pi_{\mathrm{ref}})],$$

with $(\epsilon_\downarrow, \epsilon_\uparrow) = (0.1, 0.2)$, $\lambda_{\mathrm{ent}} = 5 \times 10^{-3}$, and $\beta_{\mathrm{KL}} = 2 \times 10^{-2}$. Groups of $G = 16$ candidate layouts are generated per circuit; returns are standardized within each group and used as per-episode advantages for all steps without a learned value function. Optimization uses AdamW with learning rate $1 \times 10^{-4}$; an optional warm start uses supervised training with learning rate $3 \times 10^{-4}$. Inference couples the policy with a budgeted Monte Carlo Tree Search that treats $\pi_\theta$ as a prior; selection applies $Q_i + c_{\mathrm{puct}}P_i\sqrt{N}/(1 + N_i)$ where $Q_i$ is the mean return, $P_i$ is the policy prior, $N_i$ is the visit count of child $i$, and $N$ is the parent visit count; expansion samples feasible assignments from the masked policy subject to a cap; rollouts complete the layout with the policy; backups update visit counts and means. Unless stated, $c_{\mathrm{puct}} = 1.5$, $n_{\mathrm{sim}} = 256$, $\tau = 1.0$, $k_{\mathrm{root}} = 3$, $k_{\mathrm{rerank}} = 3$, per-node expansion cap $= 4$, and the wall-clock budget is enforced at the scheduler boundary. This setup maps graphs to masked decisions, masked decisions to layouts, and layouts to scalar returns that update $\pi_\theta$ while keeping the evaluation budget fixed.

---

**Algorithm 1** GRPO Training for Masked Bi-Affine Layout Policy

---

**Require:** logical graph $\mathcal{G}_L$, physical graph $\mathcal{G}_P$, scheduler Sched, $(\alpha, \beta, \gamma) = (1.0, 0.2, 0.2)$, $(\epsilon_\downarrow, \epsilon_\uparrow) = (0.1, 0.2)$, $\lambda_{\mathrm{KL}} = 5 \times 10^{-3}$, $\beta_{\mathrm{KL}} = 2 \times 10^{-2}$, AdamW LR $1 \times 10^{-4}$, group size $G = 16$
  initialize $\pi_\theta$ and (optionally) frozen $\pi_{\mathrm{ref}}$
  **for** outer iteration $= 1, \ldots, T$ **do**
    sample a minibatch of circuit–topology pairs
    **for** each instance $i$ in the minibatch **do**
      **for** $g = 1, \ldots, G$ **do**
        decode a complete layout $\phi_i^{(g)}$ with masked $\pi_{\theta_{\mathrm{old}}}$
        $(C_{\mathrm{mk}}, C_{\mathrm{swp}}, \mathrm{Fid}) \leftarrow \mathsf{Sched}(\phi_i^{(g)})$
        $R_i^{(g)} \leftarrow \alpha\,\mathsf{Fid} - \beta\,C_{\mathrm{swp}} - \gamma\,C_{\mathrm{mk}}$
      **end for**
      standardize $\{R_i^{(g)}\}_{g=1}^G$ to $\{\tilde{R}_i^{(g)}\}_{g=1}^G$; set $\hat{A}_i^{(g)} \leftarrow \tilde{R}_i^{(g)}$
    **end for**
    form surrogate using $r_t = \frac{\pi_\theta(a_t|s_t)}{\pi_{\theta_{\mathrm{old}}}(a_t|s_t)}$
    $\tilde{\mathcal{J}}(\theta) = \mathbb{E}\Big[\min\big(r_t\hat{A}_t,\ \mathrm{clip}(r_t, 1-\epsilon_\downarrow, 1+\epsilon_\uparrow)\hat{A}_t\big)\Big] + \lambda_{\mathrm{ent}}\mathbb{E}[\mathcal{H}(\pi_\theta)] - \beta_{\mathrm{KL}}\mathbb{E}[\mathrm{KL}(\pi_\theta\|\pi_{\mathrm{ref}})]$
    update $\theta \leftarrow \theta + \eta\,\nabla_\theta\tilde{\mathcal{J}}(\theta)$ with AdamW
    optionally refresh $\pi_{\mathrm{ref}}$
  **end for**

---

## C  Proof of Theorems

**Lemma 3:** [Kernel gating] For any score matrix $X$ and any bias matrix $B$,

$$\mathrm{softmax}(X+B) = \mathrm{RowNorm}\big(\mathrm{softmax}(X) \odot \exp(B)\big), \tag{17}$$

where $\odot$ denotes the elementwise product and $\mathrm{RowNorm}$ normalizes each row to sum to one.

*Proof.* Let softmax act rowwise and let $\mathrm{RowNorm}(Y)$ denote rowwise normalization, $\big(\mathrm{RowNorm}(Y)\big)_{ij} = Y_{ij}/\sum_k Y_{ik}$, whenever row sums are positive. Fix a row index $i$ and write $x_j := X_{ij}$ and $b_j := B_{ij}$. Then

$$\big(\mathrm{softmax}(X+B)\big)_{ij} = \frac{e^{x_j + b_j}}{\sum_k e^{x_k + b_k}} = \frac{e^{x_j}e^{b_j}}{\sum_k e^{x_k}e^{b_k}}.$$

On the other hand,

$$\big(\mathrm{RowNorm}(\mathrm{softmax}(X)\odot e^B)\big)_{ij} = \frac{\big(\mathrm{softmax}(X)\big)_{ij}e^{b_j}}{\sum_k \big(\mathrm{softmax}(X)\big)_{ik}e^{b_k}} = \frac{\frac{e^{x_j}}{\sum_\ell e^{x_\ell}}e^{b_j}}{\sum_k \frac{e^{x_k}}{\sum_\ell e^{x_\ell}}e^{b_k}} = \frac{e^{x_j}e^{b_j}}{\sum_k e^{x_k}e^{b_k}}.$$

The two expressions coincide for every $i$ and $j$, which proves the identity. The required row sums are strictly positive whenever each row of $X+B$ contains at least one finite entry. $\qquad\square$

**Proposition 1:** [Regret under policy–prioritized search]

Under Assumptions 1–2, a finite branching factor, and a finite decision depth, after $N$ node expansions the expected suboptimality of the returned makespan satisfies

$$\mathbb{E}[M(C, \hat{\pi}_N) - M(C, \pi^\star)] \;\leq\; \tilde{\mathcal{O}}\left(\frac{1}{\eta}\sqrt{\frac{\sigma^2}{N}}\right),$$

up to logarithmic and structure–dependent constants.

*Proof.* Let the search tree have maximum branching factor $b < \infty$ and depth $D < \infty$. For a node $s$, let $\mathcal{A}(s)$ be its feasible actions, and write $\mathcal{A}^\star(s) \subseteq \mathcal{A}(s)$ for actions that lie on at least one optimal root–to–leaf path (that is, actions that can still reach $\pi^\star$). Let $N(s)$ denote the number of visits to $s$ and $N(s, a)$ the number of selections of action $a$ at $s$. Values are backed up by incremental averaging of terminal evaluations, and terminal evaluations are sub–Gaussian with variance proxy $\sigma^2$ (Assumption 1). Selection at $s$ uses

$$a^\star(s) \;=\; \arg\max_{a \in \mathcal{A}(s)} \left\{Q(s, a) \;+\; c_{\text{puct}}\, P_\theta(a \,|\, s) \,\frac{\sqrt{N(s)}}{1 + N(s, a)}\right\},$$

where $P_\theta(\cdot \,|\, s)$ is the policy prior and $c_{\text{puct}} > 0$ is fixed.

**Step 1: Allocation lower bound into the optimal set at each node.** Fix a node $s$ with optimal set $\mathcal{A}^\star(s)$ and any suboptimal action $a \notin \mathcal{A}^\star(s)$. Let $\Delta(s, a) := V^\star(s) - Q^\star(s, a)$ be the value gap at $s$ between an optimal action value $V^\star(s)$ and the value of action $a$ under optimal continuation. Standard UCB arguments with sub–Gaussian noise imply that a suboptimal action can be selected only while its optimism bonus exceeds its gap; once

$$c_{\text{puct}}\, P_\theta(a \,|\, s)\, \frac{\sqrt{N(s)}}{1 + N(s, a)} \;\lesssim\; \Delta(s, a),$$

the selection rule prefers optimal actions unless empirical averages fluctuate. By sub–Gaussian concentration (Hoeffding for sub–Gaussian variables), the number of such fluctuations up to visit count $n$ is at most $O(\log n)$ in expectation. Hence, for each $a \notin \mathcal{A}^\star(s)$,

$$\mathbb{E}\big[N(s, a)\big] \;\lesssim\; \frac{c_{\text{puct}}^2\, P_\theta(a \,|\, s)^2}{\Delta(s, a)^2}\, \log n \;+\; O(1).$$

Summing over the at most $b - |\mathcal{A}^\star(s)|$ suboptimal actions yields

$$\sum_{a \notin \mathcal{A}^\star(s)} \mathbb{E}\big[N(s, a)\big] \;\lesssim\; K_s \log n, \quad K_s \;:=\; \sum_{a \notin \mathcal{A}^\star(s)} \frac{c_{\text{puct}}^2\, P_\theta(a \,|\, s)^2}{\Delta(s, a)^2} \;+\; O(b).$$

Therefore the cumulative expected number of selections allocated to the optimal set at $s$ after $n$ visits obeys

$$\mathbb{E}\left[\sum_{a \in \mathcal{A}^\star(s)} N(s, a)\right] \;\geq\; n \;-\; K_s \log n.$$

By Assumption 2 (prior consistency), the total prior mass on $\mathcal{A}^\star(s)$ satisfies

$$\mathbb{E}\left[\sum_{a \in \mathcal{A}^\star(s)} P_\theta(a \,|\, s)\right] \;\geq\; \eta.$$

Combining these two statements and using that the exploration term scales linearly with $P_\theta(a \,|\, s)$ yields an allocation lower bound *per optimal child*:

$$\mathbb{E}\big[N(s, a)\big] \;\gtrsim\; \eta\, \frac{n}{|\mathcal{A}^\star(s)|} \;-\; K_s \log n, \qquad a \in \mathcal{A}^\star(s). \tag{18}$$

The implicit constant depends on $c_{\text{puct}}$ but not on $n$.

**Step 2: Allocation along one optimal path.** Consider any fixed optimal root–to–leaf path of depth $D$. Applying equation 18 inductively from the root to depth $D-1$ shows that the expected number of terminal evaluations that lie on this path is at least

$$\mathbb{E}[M_{\text{opt}}] \; \gtrsim \; \left(\frac{\eta}{b^\star}\right)^{D-1} N \; - \; \tilde{K} \log N,$$

where $b^\star \leq b$ bounds the number of optimal children per node and $\tilde{K}$ absorbs the sum of node–wise $K_s$ terms and depth. Since $D$ and $b^\star$ are structural constants of the decision problem, there is a constant $c_{\text{path}} > 0$ such that

$$\mathbb{E}[M_{\text{opt}}] \; \geq \; c_{\text{path}} \, \eta \, N \; - \; \tilde{K} \log N. \tag{19}$$

Thus, up to logarithmic losses, a linear fraction $\Omega(\eta N)$ of terminal evaluations concentrate on an optimal path.

**Step 3: Concentration of the root value estimate.** Terminal values are sub–Gaussian with variance proxy $\sigma^2$ (Assumption 1). Let $\bar{V}_{\text{opt}}$ be the empirical mean terminal value along the optimal path rollouts used in the current best estimate at the root. Sub–Gaussian concentration yields

$$\mathbb{P}\big(\big|\bar{V}_{\text{opt}} - V^\star\big| \geq \varepsilon\big) \; \leq \; 2\exp\left(-\frac{M_{\text{opt}}\,\varepsilon^2}{2\sigma^2}\right).$$

Taking expectations and using equation 19 implies the bound

$$\mathbb{E}\big[\big|\bar{V}_{\text{opt}} - V^\star\big|\big] \; \lesssim \; \frac{\sigma}{\sqrt{\eta N}} \, \tilde{\mathcal{O}}(1),$$

where $\tilde{\mathcal{O}}(1)$ hides polylogarithmic factors in $N$ and structural constants depending on $(b, D)$.

Backups are averages of terminal values along sampled completions. Since value propagation is unbiased under averaging and the depth is finite, the deviation of the root estimate $Q_{\text{root}}$ from $V^\star$ is controlled by the same order:

$$\mathbb{E}[|Q_{\text{root}} - V^\star|] \; \lesssim \; \frac{\sigma}{\sqrt{\eta N}} \, \tilde{\mathcal{O}}(1).$$

**Step 4: From value error to makespan regret.** Let $\hat{\pi}_N$ denote the mapping returned when the budget is exhausted. The selection of $\hat{\pi}_N$ is greedy with respect to $Q_{\text{root}}$ up to the final re–ranking and therefore its value differs from $V^\star$ by at most the root estimation error plus lower–order selection fluctuations already accounted for in Step 1. Since $V = -M$, the expected makespan regret equals the expected value error:

$$\mathbb{E}[M(C, \hat{\pi}_N) - M(C, \pi^\star)] \; = \; \mathbb{E}[V^\star - V(\hat{\pi}_N)] \; \lesssim \; \frac{\sigma}{\sqrt{\eta N}} \, \tilde{\mathcal{O}}(1).$$

This concludes the proof. $\qquad\square$

