# OpenReview forum: "GRIT: Graph-Biased Transformers for Hardware-Aware Quantum Layout via Reinforcement Learning Search"
_ICLR.cc/2026/Conference — ICLR 2026 Conference Withdrawn Submission_

### Official Review · Reviewer_3xvt · 2025-10-18

**Soundness:** 2
**Presentation:** 1
**Contribution:** 2
**Rating:** 2
**Confidence:** 3

**Summary:**

This paper introduces GRIT, a hardware-aware quantum compilation framework based on RL. It proposes to use a graph-bias transformer architecture to jointly encode the logical graph and the physical graph, through modifications including Laplacian positional encodings and geodesic bias. Then, it uses GRPO to optimize the networks with receiving reward signals from a blocking-aware simulator to align with deployment-time performance. During inference time, it integrates the learned policy with MCTS to search for promising mapping solutions.

**Strengths:**

- Routing quantum circuits on hardware devices is an important problem. Using RL to address this search problem is a promising solution.
- The background knowledges are well introduced in the paper.
- The idea of jointly modeling two graphs sounds novel.

**Weaknesses:**

- This paper seems to be incomplete. There is no related work in the main body to justify the novelty and clarify the connections with previous works. The section of experiments and results takes only less than one page.
- The use of Laplacian positional encoding has no citation, though it is easy to find relevant papers on the Internet.
- The experimental results show little improvement brought by the proposed method. The effectiveness of the whole framework is not empirically justified.
- There is no ablation study to justify various design choices.

**Questions:**

Please see the weaknesses section above.

---

### Official Review · Reviewer_tBDN · 2025-10-31

**Soundness:** 2
**Presentation:** 1
**Contribution:** 1
**Rating:** 2
**Confidence:** 4

**Summary:**

This paper proposes GRIT, a framework for quantum qubit mapping and routing that combines a novel "Graph-Biased Transformer" (GBT) architecture with reinforcement learning (RL) and Monte Carlo Tree Search (MCTS). The authors claim that this hybrid approach addresses key limitations in prior work, such as "objective mismatch" (by training with a hardware-aware simulator) and "representation limits" (by using the GBT to jointly encode logical and physical graphs). The framework uses the GBT, trained via RL, as a policy prior to guide an MCTS planner at inference time. The paper presents experiments on various benchmarks and hardware backends, claiming improved fidelity and efficiency over Qiskit's default transpiler and a random baseline.

**Strengths:**

1. Strong Problem Formulation: The paper correctly identifies a critical and practical challenge in near-term quantum computing. The diagnosis of the "objective mismatch" (i.e., optimizing proxy metrics like SWAP count instead of true hardware performance like makespan) is accurate and provides a strong motivation for the work [cite: 2635-2636].

2. Well-Motivated High-Level Framework: The high-level design of the proposed solution is logical. The idea of using a learned policy, trained with a realistic hardware simulator (Fig 1b), to guide a robust search algorithm like MCTS (Sec 4.2) is a powerful and well-established paradigm for complex planning problems.

**Weaknesses:**

Despite its sound motivation, the paper is plagued by severe issues in its organization, theoretical justification, and experimental validation, which ultimately fail to support its central claims.

1. The paper is poorly organized. It dedicates nearly five pages to introductory and background material, leaving the core model (Sec 3) and learning (Sec 4) sections compressed, underdeveloped, and vague. Critical details of the architecture and training are hard to parse, while extensive space is given to standard definitions.

2. The paper attempts to bolster its contribution by including formal-looking theoretical analysis (e.g., Lemma 3, Proposition 1) that is ultimately superficial. Lemma 3 is a trivial algebraic identity, and Proposition 1 is a standard (and unsurprising) regret bound for policy-guided MCTS. This "theory" provides no novel insight into the specific architectural innovations of the Graph-Biased Transformer and feels disconnected from the paper's core claims.

3. The paper's primary claim is its complex GBT architecture, yet this claim is left entirely unsubstantiated by the experiments. There are zero ablation studies to demonstrate that the novel components (Laplacian PE, Geodesic Bias) offer any advantage over a standard Transformer or a simpler GNN. Furthermore, the baselines used ("RANDOM" and "QISKIT TRANSPILER") are critically insufficient. Qubit routing is a mature field, and failing to compare against any SOTA heuristic (e.g., SABRE) or other ML-based compilers makes it impossible to assess the method's actual contribution.

4. The experimental results fail to justify the model's complexity, even against the weak baselines. As shown in Tables 1 & 2, GRIT often produces worse results (e.g., higher Depth on QFT, more SWAPs on 127q QFT) than the simple Qiskit heuristic. The runtime (TIME(s)) comparison is fundamentally flawed: the baselines are fast, constant-time heuristics (~20-30s), while GRIT is a time-budgeted MCTS search. The results show GRIT is slower, scales poorly, and still fails to deliver a consistent win on the key metrics, demonstrating no practical value for its complex design.

**Questions:**

1. The experimental results (e.g., QFT-127q in Table 2) show that GRIT is significantly slower than the Qiskit heuristic (120s+ vs. 72.84s) and produces a worse-quality circuit (4113 SWAPs vs. 4078). How do the authors reconcile this negative result with their claims of "improved fidelity and efficiency"?

---

### Official Review · Reviewer_unKn · 2025-10-31

**Soundness:** 3
**Presentation:** 2
**Contribution:** 2
**Rating:** 4
**Confidence:** 3

**Summary:**

This work presents a reinforcement-learning-based, hardware-aware quantum circuit compilation algorithm. The method uses a transformer to simultaneously process logical and physical qubit graphs with a structural bias and is trained using a policy gradient with hardware constraints. Numerical results on a Qiskit simulator demonstrate that the proposed method achieves a smaller number of SWAP operations and shallower circuit depth in most cases.

**Strengths:**

This work presents a novel framework for quantum compiling tailored for NISQ devices.

**Weaknesses:**

- The manuscript is not well-organized. For example, the Background section occupies an excessively large portion of the main text.

- The proposed method encounters scalability issues for large-scale systems, as the action sequence length grows with the system size, requiring more computation for the search process. This limitation is corroborated by the long compilation times reported for the 127-qubit devices in Tables 1 and 2.

**Questions:**

See the Weakness section

---

### Official Review · Reviewer_xTHp · 2025-11-01

**Soundness:** 3
**Presentation:** 3
**Contribution:** 2
**Rating:** 2
**Confidence:** 3

**Summary:**

Overall, the approach is well-motivated and clearly articulated. However, the results do not indicate that the approach is successful in reducing SWAPs or circuit depth. On smaller qubit counts, the compile time is significantly reduced with respect to the comparison methods, but compilation time grows quite large for larger numbers of qubits. Further evaluation is needed to see which parts of this novel approach are beneficial and where further improvements can be made to reduce the search time with larger qubit counts.

**Strengths:**

Clearly written motivation explaining the challenges of mapping and routing in quantum computing as combinatoric problems. Finding a machine learning approximation would impact the practicality of quantum computing especially for solving larger problems. Compared with traditional approaches, a machine-learning approach should be more flexible for generalizing to different hardware and for diverse circuits.

**Weaknesses:**

It’s not clear that the results are an improvement over the comparison methods, despite the novel approach. An ablation study would help to analyze how much each part of the workflow (graph-biased transformer or RL-guided search or using makespan for the objective, etc) is contributing to the performance.

It is concerning that the compile time grows so much with qubit count. It is most pronounced with QFT, but the growth in compile time also exists for QAOA Ring and GHZ. More analysis is needed to see if the suggested approaches to limit the search space will reduce the time while maintaining the reduction in swaps and depth.

In Table 1, under 127Q and QFT, depth of 5312 should be bold instead of 5341.

**Questions:**

Can you test how well the combined graph representation is working? These are complex interactions and so it is questionable whether they can be learned.

Similarly, how could you test whether the use of makespan in the objective is an improvement over proxies like SWAPs and depth? Your performance metrics in the Tables are only SWAPs and depth, so these do not seem to be proxies.

---

### Note · Authors · 2025-12-19

I have read and agree with the venue's withdrawal policy on behalf of myself and my co-authors.